# Human Umbilical Cord-Based Therapeutics: Stem Cells and Blood Derivatives for Female Reproductive Medicine

**DOI:** 10.3390/ijms232415942

**Published:** 2022-12-14

**Authors:** Adolfo Rodríguez-Eguren, María Gómez-Álvarez, Emilio Francés-Herrero, Mónica Romeu, Hortensia Ferrero, Emre Seli, Irene Cervelló

**Affiliations:** 1IVI Foundation, Health Research Institute La Fe, 46026 Valencia, Spain; 2Department of Obstetrics, Gynecology and Reproductive Sciences, Yale School of Medicine, New Haven, CT 05610, USA; 3Department of Pediatrics, Obstetrics and Gynecology, School of Medicine, University of Valencia, 46010 Valencia, Spain; 4Gynecological Service, Consortium General University Hospital of Valencia, 46014 Valencia, Spain; 5IVIRMA New Jersey, Basking Ridge, NJ 07920, USA

**Keywords:** umbilical cord, female reproduction, umbilical cord mesenchymal stem cells, stem cell therapy, platelet-rich plasma, exosomes, growth factors, ovary, endometrium

## Abstract

There are several conditions that lead to female infertility, where traditional or conventional treatments have limited efficacy. In these challenging scenarios, stem cell (SC) therapies have been investigated as alternative treatment strategies. Human umbilical cord (hUC) mesenchymal stem cells (hUC-MSC), along with their secreted paracrine factors, extracts, and biomolecules, have emerged as promising therapeutic alternatives in regenerative medicine, due to their remarkable potential to promote anti-inflammatory and regenerative processes more efficiently than other autologous treatments. Similarly, hUC blood derivatives, such as platelet-rich plasma (PRP), or isolated plasma elements, such as growth factors, have also demonstrated potential. This literature review aims to summarize the recent therapeutic advances based on hUC-MSCs, hUC blood, and/or other plasma derivatives (e.g., extracellular vesicles, hUC-PRP, and growth factors) in the context of female reproductive medicine. We present an in-depth analysis of the principal molecules mediating tissue regeneration, compiling the application of these therapies in preclinical and clinical studies, within the context of the human reproductive tract. Despite the recent advances in bioengineering strategies that sustain delivery and amplify the scope of the therapeutic benefits, further clinical trials are required prior to the wide implementation of these alternative therapies in reproductive medicine.

## 1. Introduction

Infertility is a disease of the male or female reproductive system, defined by the World Health Organization as the failure to achieve a pregnancy, after at least 12 months of regular unprotected sexual intercourse [1]. Specifically, female infertility can be caused by various disorders of the reproductive system, including premature ovarian insufficiency (POI), polycystic ovary syndrome (PCOS), endometriosis, Asherman’s syndrome (AS), or endometrial atrophy (EA), among others.

Several preclinical studies and clinical trials have been conducted to evaluate the use of stem cells (SCs), and/or their derivatives, as an alternative strategy for treating different reproductive disorders that lead to infertility [2,3,4]. Stem cells are undifferentiated cells, with the potential to perpetuate self-renewal, for a long period of time. They can divide into identical SCs (via symmetrical division), or, under certain physiologic or experimental stimulus, give rise to differentiated (mature) cells with particular functions (via asymmetric division or tissue-specific differentiation processes) [5]. Several studies have focused on mesenchymal stem cells (MSCs) for experimental approaches to treat infertility [6,7,8,9,10,11,12,13,14,15,16]. In this context, there are various sources of human MSCs used in therapeutic approaches for regenerative medicine, such as the bone marrow, adipose tissue, menstrual blood, salivary glands, dental pulp, amniotic fluid, placental tissue, and umbilical cord (UC), the latter being the focus of this review [17,18].

Until the 1990s, the human UC and its blood derivatives were considered medical waste. However, the collection of human UC-MSCs (hUC-MSC) is considered non-invasive, and thus not encumbered with ethical problems. Remarkably, hUC-MSCs have some exceptional characteristics, including rapid self-renewal, low oncogenicity, and poor immunogenicity (due to their low expression of the major histocompatibility complex (MHC) class I and class II proteins), making them an important source of SCs for allogeneic transplantation therapy without immune rejection [19,20]. Moreover, the cell source is heterologous, its procurement is non-invasive and is not associated with any comorbidity for the patient. Considering these features, hUC-MSCs are preferentially used over other sources of MSCs, for auto- and allo-transplantation. Finally, different methods have been developed to isolate hUC-MSCs (i.e., such as from Wharton’s jelly, arteries or veins) [20,21,22,23]. To date, cell therapy based on hUC-MSCs has been applied in a multitude of medical disciplines, for regenerative and immunomodulatory purposes [20,24,25,26,27,28,29], with gynecology being no exception [30,31,32,33,34,35,36,37,38,39,40]. Nevertheless, many concerns remain regarding their use and long-term safety, and most of the treatments developed thus far are still considered invasive and experimental [41,42].

In this regard, new strategies have been investigated. In particular, the synthesis and secretion of chemokines, growth factors (GFs), blood-extracts, biomolecules, and hormones by hUC-MSCs affects the adjacent cells through paracrine signaling. These components play important roles in angiogenesis, anti-inflammation, immunomodulation, anti-apoptosis, and anti-fibrosis, thereby contributing to the regeneration of injured tissues [43]. For example, microvesicles and exosomes are extracellular vesicles (EVs) that carry membrane and cytoplasmic constituents of intracellular origin, and can transfer proteins, messenger RNAs, microRNAs, and bioactive lipids to target cells (with specific surface receptors) to affect their phenotype and function [44,45]. Based on this premise, some groups have explored using hUC-MSC-derived conditioned media, and hUC-MSC-secreted vesicles, rather than the hUC-MSCs directly, to examine the indirect therapeutic effects on damaged tissues [43,46]. Notably, the microvesicles and exosomes released from MSCs exerted a protective effect on reproductive tissues, and rescued ovarian function, in vitro and in vivo [47,48,49].

Overall, hUC offers a plethora of valuable byproducts for regenerative medicine, including whole hUC blood and its derivatives (i.e., serum, plasma, endothelial progenitor cells, MSCs, and hematopoietic cells). Although hUC blood is mainly used to treat blood disorders via hematopoietic SC transplantation, hUC-MSCs have garnered the most attention for therapeutic purposes [50,51]. Recently, platelet-rich plasma (PRP), a blood-derived product that can be prepared commercially and contains five to ten times the density of platelets than normal blood [52], has also emerged as a promising regenerative mediator. Interestingly, studies analyzing human umbilical cord platelet-rich plasma (hUC-PRP) reported that it produced a higher concentration of GFs, chemokines, and cytokines than adult peripheral blood PRP, and contributed to tissue regeneration through enhancement of vascular endothelial growth factor (VEGF) and platelet-derived growth factor (PDGF) [53,54,55,56]. So far, hUC-PRP has shown promising preclinical results, for epidermolysis bullosa [57,58], oral mucositis [59], and foot ulcers associated with diabetes [60], and reversing chemotherapy-induced ovarian damage [61].

Overall, UC blood offers numerous biological agents/products with high regenerative and immunomodulatory potential. This bibliographic review aims to compile the most relevant studies in the context of female reproduction, highlighting the advances and the emerging therapeutic strategies.

## 2. Human Umbilical Cord: Composition and Potential Mechanisms of Action

### 2.1. The Cellular Components

The hUC blood contains different cell types, including hUC-derived unrestricted somatic SCs, embryonic-like SCs, multipotent progenitor cells, and endothelial progenitor cells [62]. However, the most studied are the hUC-MSCs, which exert potent immunosuppressive and anti-inflammatory effects, as demonstrated with cell-based therapies for heart injury [63,64], retinitis pigmentosa [65], type II diabetes [66], and female pathologies [67,68], among others.

All these cellular components, particularly hUC-MSCs, utilize three processes during cell therapy (Figure 1A): (i) homing (the process of migrating to and settling in the damaged tissue), (ii) secretion of GFs and EVs, and (iii) immunomodulation. Briefly, hUC-MSCs colonize the injury site, and participate in cell proliferation and angiogenesis, through the release of paracrine factors (e.g., cytokines, GFs, and EVs), while simultaneously modulating the immune system to control inflammation and promote tissue regeneration [62]. To elucidate the complexities of these processes, each one is described in detail below, to elaborate on relevant cell functions and/or hUC blood factors.

#### 2.1.1. The Homing Process

When tissue integrity is compromised, the surrounding cells release inflammatory molecules and chemokines, producing a concentration gradient of ligands that recruits innate immune cells, via a phenomenon known as chemotaxis. Stem cells are attracted in response to this chemical stimulus and migrate directly to the site of damage to initiate specific regenerative functions. This homing effect is made possible by the adhesive junctions between SCs and the vascular endothelium at the target tissue. Specifically, the cell-rolling interactions are mediated by the homing receptors expressed on the SCs and their cognate endothelial co-receptors. The activation of integrins anchor the engrafted cells for extravasation [62,69]. To achieve this cell mobilization, MSCs express the most common immune-cell-homing chemokine receptors, including C-X-C chemokine receptor type 4 (CXCR4) and C-C chemokine receptor (CCR), which interact with the potent chemoattractants that are upregulated at the injury site, such as stromal-cell-derived factor 1 (SDF1) and monocyte-chemotactic protein 3 (MCP3) [70,71].

#### 2.1.2. Secretion of Paracrine Factors

After homing, SCs secrete cytokines, biomolecules, trophic and growth factors. These molecules impact target cells by modulating inflammation/apoptosis, triggering progenitor cell proliferation, and stimulating tissue repair to provide favorable conditions for cell survival. Indeed, GFs and EVs are regarded as the two main components implicated in SC paracrine signaling [62].

##### Growth Factors

Hundreds of GFs (including VEGF, nerve growth factor (NGF), endothelial growth factor (EGF), fibroblast growth factor (FGF), hepatocyte growth factor (HGF), and transforming growth factor beta (TGFβ)) and cytokines (such as interleukins (ILs), granulocyte colony-stimulating factor (G-CSF), and granulocyte–macrophage colony-stimulating factor (GM-CSF)) are secreted by hUC-MSCs, after fusion of secretory granules with the plasma membrane, to immediately begin coordinating local tissue regeneration [72,73] (Table 1). These factors interact with membrane receptors on adjacent recipient cells [62], to mediate angiogenesis (in particular VEGF), cell chemotaxis (mainly driven by specific chemokines), anti-apoptotic effects (particularly G-CSF and NGF), proliferation and immunomodulation [74]. Interestingly, adult bone marrow MSCs and hUC-MSCs have comparable cytokine expression profiles, which translate into similar biological potential [75].

The paracrine action of the hUC-MSCs has previously been described in various pathologies, such as renal injury [76,77,78], brain damage [79,80], cardiac injury [64,81], and female reproductive pathologies. For example, G-CSF has shown protective effects in ovarian damage by activating the PI3K/AKT pathway to promote cell survival and proliferation of primordial follicles and granulosa cells [82,83]. Notably, this signaling pathway is co-activated by NGF, which, in turn, signals through the tropomyosin receptor kinase A (TrkA) receptor, contributing to these downstream cellular responses [84]. Alternatively, TGFβ binds to its cognate receptors on the oocyte surface and inhibits follicular cell apoptosis in ovarian failure [85]. Taken together, these findings suggest that each GF modulates a particular molecular pathway, contributing to a complex network of cell processes, that ultimately produce a tailored response in the target tissue. Remarkably, many of these factors appear to act synergistically [84], which partially justifies the difficulty of elucidating the individual roles of each GF secreted by hUC-MSCs.

##### Extracellular Vesicles

The therapeutic benefits produced by the EVs derived from hUC-MSCs depend on their mechanism of action and the pathophysiology of each disease. For example, in rats with acute kidney injury, treatment with EVs downregulated pathways activated by p38 mitogen-activated protein kinase (MAPK), which consequently decreased the caspase 3 protein expression and favored renal cell survival. In this context, EVs also promoted cell proliferation, through activation of the extracellular-signal-regulated kinase (ERK) 1/2 pathway, reducing creatinine levels, kidney tubule necrosis, apoptosis, and oxidative stress in vivo [86]. Meanwhile, treating liver fibrosis (characterized by the excessive production and deposit of collagen resulting from tissue repair process) with EVs reverted the conversion of hepatocytes to fibroblasts (that occurs through the epithelial-to-mesenchymal transition (EMT)), both in vitro and in vivo. Specifically, the molecules carried by the EVs induced downregulation of TGFβ, which inhibited SMAD family member 2 (Smad2) phosphorylation, that is essential for the transcription of genes responsible for the EMT [87]. Interestingly, female disorders such as intrauterine adhesions [88] or PCOS [89] also benefited from EV treatment both in vivo and in vitro. In both these cases, the therapeutic effect of the EVs was directed by the post-transcriptional modifications produced by the micro-RNAs, that induced M2 macrophage polarization and inhibited the expression of pro-inflammatory molecules [88,89].

While small non-coding RNAs are usually transported by EVs, they can also be secreted directly by the hUC-MSCs to modulate other signaling cascades [90,91,92,93,94]. Indeed, the upregulation of miR-455-5p and miR-330-5p in hUC-MSCs has been associated with different mechanisms that attenuate endometrial injury and promote repair of murine damaged endometrium. In particular, miR-455-5p regulates the suppressor of cytokine signaling 3 (SOCS3)-mediated Janus kinase (JAK)/signal transducer and activator of transcription 3 (STAT3) signaling pathway [90], whereas miR-330-5p stabilizes mitochondrial metabolism through pyruvate dehydrogenase kinase 2 (PDK2) signaling [93].

#### 2.1.3. Immunomodulation

Since the hUC-MSCs scarcely express MHC class II proteins and co-stimulatory molecules, they are considered immunoprivileged cells [29]. These cells can suppress the immune system by inhibiting the physiological functions of T-, B-, and natural killer (NK) cells (e.g., cell proliferation, production of GFs, and cytotoxicity), through cell–cell contact, and the secretion of paracrine factors. Specifically, during adaptive immune responses, hUC-MSCs restrained the activation and proliferation of cluster of differentiation (CD) 4 (CD4)+, CD8+, CD2+, and CD3+ T-cells subpopulations, while stimulating regulatory T-cells [95]. In addition, hUC-MSCs significantly suppressed the proliferation, differentiation, and immunoglobulin secretion of B-cells, in vitro, by modulating the MAPK signaling pathway [96]. Regarding the innate immune response, hUC-MSCs suppressed NK cells by secreting prostaglandin-E2 [97], and converted monocytes and dendritic cells to an immature state [29]. Although hUC-MSC-mediated immunosuppression has been described in many studies [29,95,96,98,99], its complexities merit further investigation. Furthermore, the hUC-MSCs’ immune modulating effects will notably depend on the ratio of hUC-MSCs and immune cells (especially B-cells), immune cell activation state and maturation stage. In this regard, the microenvironment is crucial for determining the immunomodulatory effects of the hUC-MSCs [96].

Notably, recent studies proposed using the factors secreted by hUC-SCs, rather than cell transplantation, for regenerative medicine [100,101,102,103]. The secreted GFs are referred to as the secretome of the hUC-SCs, can be isolated from culture media, and induce biological effects similar to SC therapy. The advantages of using this conditioned culture medium include that it can be manufactured more easily, is acellular, immunotolerated, and does not promote tumorigenesis [104]. On the other hand, several authors have isolated EVs by ultracentrifugation [105] and studied the specific effects of SC-derived EVs [81,87,89,106,107,108,109,110], including the re-epithelialization of cutaneous wounds [111], muscle regeneration [112], and modulation of immune system [113], among others. Nevertheless, it is still necessary to compare the efficacy and potency of conditioned media and EV-based therapies with SC transplantation, in controlled clinical trials.

### 2.2. The Acellular Fractions

The postulate that tissue repair by hUC-MSCs is mediated via paracrine signaling suggests that the GFs and cytokines these cells produce are sufficient to activate their exceptional regeneration abilities and provides an alternative approach for using hUC blood derivatives in regenerative medicine.

In this context, recent studies have focused on using the acellular fractions of the hUC blood, namely the human umbilical cord serum (hUCS) and plasma (hUCP) (Figure 1B). Since their GFs and cytokines produced comparable anti-inflammatory, angiogenic, and anti-apoptotic effects, the main difference between both biofluids is the absence (hUCS) or presence (hUCP) of clotting factors [114,115]. In addition, since hUCS and hUCP resemble the ‘culture medium’ of hUC blood cells, it is possible that they contain some GFs secreted by hUC-MSCs, and therefore partly contribute to tissue regeneration processes [104].

#### 2.2.1. Human Umbilical Cord Serum

The hUCS is the non-cellular supernatant obtained when whole hUC blood clots, and is rich in cytokines and GFs with relevant biological properties, such as many ILs, G-CSF, GM-CSF, FGF, NGF, tumor necrosis factor beta, and VEGF, among others [114,115] (Table 1). Compared to hUCP, the hUCS is distinguished by a higher concentration of VEGF, which has been associated with pronounced angiogenic and anti-apoptotic effects, via PI3K/AKT and MAPK signaling pathways. However, the elevated secretory activity of the hUC blood cells, contributing to the composition of the hUCS during clotting, may also lead to this concentration difference [115].

The hUCS has been utilized as a substitute for fetal bovine serum, due to its equal or even greater culture potential with lower immunogenicity [114]. In fact, many studies have demonstrated its versatility for expanding cultures of different cell types, ranging from multipotent mesenchymal stromal cells [116] to epithelial cells [117,118] and oocytes for assisted reproductive techniques [119]. Furthermore, hUCS has shown remarkable efficacy in experimental and clinical settings, particularly in ocular injury [120,121,122,123,124]. In these cases, it provided neuroprotective action on the optic nerve, by significantly decreasing the presence of pro-inflammatory cytokines [125].

#### 2.2.2. Human Umbilical Cord Plasma

In contrast to hUCS, isolating hUCP requires the addition of an anticoagulant to prevent the platelet-mediated clotting process. hUCP is also enriched with powerful stimulators of regeneration and resident progenitor cells, however its concentration of pro-inflammatory cytokines is minimal [115]. Following centrifugation, the platelet content of hUCP can be concentrated up to 4–5 times to obtain hUC-PRP. Once activated, platelets change shape and release the contents of their alpha granules into their surroundings to chemoattract host cells that repair the tissue injury [126,127]. Next, platelets secrete specific proteins (e.g., PDGF, TGFβ, VEGF, and EGF) that initiate intracellular signaling cascades which produce specific cell responses (Table 1). Target cells include fibroblasts, bone marrow-derived SCs, and pre-osteoblasts (respectively, representing long-term healing, bone regeneration, and bone remodeling) along with immune cells (i.e., macrophages, which, in response to hUC-PRP, have decreased inflammation in numerous diseases [128]). Remarkably, while adult human PRP has been used extensively in many therapeutic areas [129,130,131,132,133,134,135,136], clinical application of hUC-PRP remains experimental [137,138,139].
ijms-23-15942-t001_Table 1Table 1Main growth factors secreted by hUC-MSCs and found in the acellular components of hUC blood. For each growth factor, this table lists the most prominent signaling pathways, biological functions, and therapeutic applications reported in the literature in reproductive medicine and other medical fields.Type of FactorSignaling PathwaysBiological FunctionsTherapeutic ApplicationsReferencesVEGFPI3K/AKTRas/MAPKsSrc/FAK✓Angiogenesis;✓Vessel permeability;✓Cell proliferation;✓Cell motility;✓Immune cell chemotaxis.Reproductive medicine: EA, IUA, POIOther fields: Neurodegenerative diseases, chronic diabetic wounds, neuropathic pain[54,110,139,140]NGFPI3K/AKTRas/MAPKsPLC-ϒJNK✓Growth and survival of sensory nerves;✓Inhibition of apoptosis;✓Cell proliferation.Reproductive medicine: POIOther fields: Brain and nerve injury, myocardial infarction, diabetic cystopathy[84,141,142,143,144]EGFPI3K/AKTRas/MAPKsJAK/STATs✓Mitosis, differentiation, and chemotaxis in epithelial and mesenchymal cells;✓Secretion of cytokines.Reproductive medicine: IUA, POIOther fields: Dry eye syndrome, atopic dermatitis[124,145]FGFPI3K/AKTRas/MAPKs✓Proliferation, growth, and differentiation of mesenchymal cells, chondrocytes, and osteoblasts;✓Angiogenesis.Reproductive medicine: EA, IUA, vaginal reconstructionOther fields: Autoimmune encephalitis, chronic diabetic wounds, amyotrophic sclerosis, osteoarthritis[144,146,147,148]HGFPI3K/AKTRas/MAPKsJAK/STATs✓Regulation of cell migration;✓Cell growth;✓Fibrosis inhibition;✓Regulation of wound healing;✓Immune modulation.Reproductive medicine: POIOther fields: Parkinson’s disease, cardiopathies, liver fibrosis[149,150,151,152]G-CSFPI3K/AKTRas/MAPKsJAK/STATs✓Cell survival and proliferation;✓Enhancement of mature neutrophil function.Reproductive medicine: POI, recurrent implantation failureOther fields: Neurodegenerative diseases, acute liver failure, brain injury[54,83,153,154,155]GM-CSFPI3K/AKTRas/MAPKsJAK/STATs✓Activation of macrophages; ✓Differentiation of immune cells; ✓Cell survival and proliferation.Reproductive medicine: POIOther fields: Lung injury[156,157,158]PDGF (*)PI3K/AKTJAK/STATsRas/MAPKsPLC-ϒ✓Mitosis and chemotaxis of mesenchymal-origin cells;✓Activation of macrophages;✓Angiogenesis and blood vessel repair.Reproductive medicine: IUAOther fields: Chronic diabetic wounds, acute kidney injury, liver fibrosis, lung diseases[159,160,161,162,163]TGFβCanonical: SMADNon-canonical:PI3K/AKTRas/MAPKs✓Collagen synthesis;✓Angiogenesis;✓Cell chemotaxis;✓Cell growth;✓Inhibition of osteoclast formation and bone resorption;✓Promotion of wound healing;✓Inhibition of apoptosis;✓Remodeling of the ECM. Reproductive medicine: IUA, POI, breast cancerOther fields: Atopic dermatitis, liver fibrosis, renal fibrosis, lung injury, wounds[87,164,165,166,167,168]ILs PI3K/AKTRas/MAPKsJAK/STATs✓Chemotaxis;✓Cell proliferation;✓Angiogenesis;✓Immune modulation.Reproductive medicine: IUA, POI, ovarian carcinomaOther fields: Autoimmune encephalitis, neuropathic pain, spondyloarthritis, brain injury, dermatitis[139,144,169,170,171,172,173]CKsPI3K/AKTRas/MAPKsJAK/STATsPLC-ϒ✓Chemotaxis and cell adhesion;✓Cell growth; ✓Cell proliferation;✓Release of granules and oxidants. Reproductive medicine: IUA, POIOther fields: Liver failure, lung injury, brain injury[94,174,175,176,177,178](*) PDGF is secreted by platelets, but it was included in the table because of its key role in acellular therapies. VEGF, vascular endothelial growth factor; NGF, nerve growth factor; EGF, endothelial growth factor; FGF, fibroblast growth factor; HGF, hepatocyte growth factor; G-CSF, granulocyte colony-stimulating factor; GM-CSF, granulocyte–macrophage colony-stimulating factor; PDGF, platelet-derived growth factor; TGFβ, transforming growth factor beta; ILs, interleukins; CKs, chemokines; PI3K, phosphoinositide 3-kinase; MAPKs, mitogen-activated protein kinases; PLC-ϒ, phospholipase C-gamma; JNK, c-Jun N-terminal kinase; ECM, extracellular matrix; IUA, intrauterine adhesions; POI, premature ovarian insufficiency.


## 3. Application of Umbilical Cord Stem Cells and Their Derivatives in the Ovary

### 3.1. Cellular Therapies Based on hUC-MSCs: Current Applications, Administration, and Fertility Restoration

Within the female reproductive tract, the ovary is the organ most commonly treated with hUC cells, especially MSCs. The main reports of the use of hUC-MSCs and UC blood derivatives were included in Table 2. In 2013, Wang et al. [35] pioneered the use of hUC-MSCs to treat POI in mice, restoring ovarian function, serum estrogen levels, adequate follicle development, and notably reducing cell apoptosis.

Similar to how other cell therapies have been infused in rodents [179], hUC-MSCs have been administered via tail vein injection, to foster their natural migration towards the injured site [84,180,181,182,183,184,185,186,187]. This strategy not only improved ovarian morphology, primordial follicle development, and hormone production, but also restored estrous cyclicity [84,181,183,186], implying ovarian performance was recovered. More importantly, systemic treatment with hUC-MSCs restored fertility in all cases [181,185,187]. Furthermore, hUC-MSC therapy provoked ovarian secretion of cytokines such as HGF, VEGF, and insulin growth factor 1 (IGF1), suggesting ovarian regeneration was mediated by paracrine mechanisms, and these factors could improve the ovarian function and delay ovarian senescence [180]. Using metabolomics, Zhao et al. [181] showed that hUC-MSCs activated the phosphoinositide 3-kinase (PI3K) pathway by stimulating synthesis of free amino acids, improved lipid metabolism, and decreased concentration of monosaccharides [181]. Meanwhile, Lu’s group [182] also focused on the protective effect of the hUC-MSCs on the theca-interstitial cells. In vitro, they observed an interesting reduction in autophagy levels in the theca-interstitial cells, induced by the decrease in oxidative stress and regulation of the AMP-activated protein kinase (AMPK)/Mammalian target of rapamycin (mTOR) signaling pathway. Augmented TrkA and NGF following hUC-MSC treatment was also reported by Zheng et al. [84], corroborating the involvement of the NGF/TrkA signaling pathway in ovarian regeneration. Moreover, the efficacy of multiple intravenous injections in a chemotherapy-induced POI mouse model was analyzed by Lv et al. [187]. After introducing cells, the ovarian morphology, follicle count, and fertility improved, regardless of whether mice received a single or triple administration. However, higher levels of serum anti-Müllerian hormone (AMH) and local Ki67 expression suggested that multiple doses of hUC-MSCs have a superior therapeutic effect than a single hUC-MSC bolus.

Alternatively, local intraovarian injections were also reported [37,188,189], and were capable of restoring follicle development and fertility in rodents. Interestingly, Shi et al. [188] analyzed the toxicity associated with injecting increasing doses of hUC-MSCs. The maximum tolerated dose was 10^6^ cells/ovary, with severe toxicity and lethality observed after exceeding this limit. Meanwhile, Pan et al. [189] found that hUC-MSC and amniotic MSC treatment, comparably restored damaged ovarian morphology, functionality, elasticity, and toughness. Finally, intraperitoneal treatment with hUC-MSCs showed similar results in terms of follicle populations, hormone regulation, and fertility restoration [190].

Despite these promising findings, administration routes for hUC-MSCs remain controversial. Three independent groups compared the efficacy of intravenous or local administration for POI models [31,36,191]. In these studies, both routes reduced cell apoptosis, augmented the proportion of growing follicles, and restored the ovarian function (as determined by regulated hormonal levels and recuperated estrous cyclicity). On the other hand, Song’s group [36] did not find any differences, Zhang et al. [191] reasoned there was a better restoration of the ovarian function after intravenous treatment, and Zhu et al. [31] found that local injection was more efficient. In terms of ovarian aging models, Zhang et al. [192] observed the positive effects of hUC-MSCs regardless of the injection route (intravenous or intraovarian), with increased follicle development, restored fertility, and reduced apoptosis of granulosa cells and reactive oxygen species (ROS) production. Finally, in humans, ovarian injection of hUC-MSCs rescued ovarian function, increased follicle development, and ultimately, led to live births [193].

Recently, Lu et al. [194] elegantly studied the impact of hUC-MSCs throughout the ovary and the endometrium. The SCs were introduced via tail injection, two weeks after inducing a POI condition. They reported improved ovarian morphology and folliculogenesis, as well as restored serum hormone levels. Regarding the endometrium, an increased number of glands and enhanced neoangiogenesis were observed, in addition to a noticeable overexpression of Homeobox A10 (HOXA10, an essential regulator of endometrial decidualization [195]), altogether revealing improved morphology and functionality. Furthermore, the type 1/2 T helper (Th1/Th2) cell ratio, and expression of NK cells, significantly decreased in the endometrium, suggesting treatment significantly regulated the ovarian function and endometrial receptivity. Likewise, Aygün et al. [196] loaded hUC-MSCs in hyaluronic acid scaffolds, and transplanted intraperitoneally in an abdominal adhesion rat model, and demonstrated the synergistic effect of the SCs on restoring hormone levels, improving follicle development and endometrial angiogenesis (after reducing the adhesions macroscopically).

While hyaluronic acid is the most popular scaffold used to develop alternative treatments in murine models of ovarian pathologies [197,198], other synthetic scaffolds have been mixed with biological products to enhance their pharmacodynamics [199]. Notably, only Sun’s group [200] has translated the use of collagen scaffolds from mice [201] to humans. Nevertheless, in mice, both hyaluronic acid and collagen scaffolds sustained the effects of the hUC-MSCs, improving ovarian morphology, cell proliferation, angiogenesis, and ultimately, restoring fertility [197,198,201]. Likewise, women treated with the hUC-MSC-collagen complex achieved a complete restoration of fertility, with primordial follicle activation (determined by the expression of phosphorylation of Forkhead box O3a (FOXO3a) and Forkhead box O1 (FOXO 1)), increased proportion of growing follicles, and consequent high serum estradiol concentration [200].

### 3.2. Emerging Alternatives: Acellular Therapies

#### 3.2.1. Extracellular Vesicles

When hUC-MSCs are isolated and applied for regenerative treatments, the bioactive executors of hUC-MSCs effects (including EVs) remain present in the cell secretome [202]. Based on this premise, independent groups have employed the medium collected from hUC-MSC culture as a reparative treatment for POI models [82,197], and found it increased AMH levels and folliculogenesis.

Specifically, several studies have described that hUC-MSCs secrete EVs, such as exosomes and microvesicles, to deliver biomolecules (i.e., lipids, carbohydrates, nucleic acids, and proteins) that facilitate cellular and host cell reprogramming [62,203]. Consistent with the findings of two other independent groups [49,107], Liu et al. [204] reported that administering EVs, derived from the hUC, through the tail vein efficiently improved fertility of mice with POI, by improving ovarian and follicle morphology, balancing hormone levels, returning estrous cyclicity, and reducing apoptosis.

The intraperitoneal transplantation of exosomes also proved beneficial in mice, by returning folliculogenesis and hormones to nearly normal levels [205]. This improvement in reproductive outcomes was associated with the regulation of the Hippo pathway, which is critical for regulating follicle activation and survival, and thus, ovarian function [206]. Furthermore, Ding et al. [48] demonstrated, in a murine model, that exosomal miRNA-17-5p was strongly associated with improvements in ovarian function. Indeed, the inhibition of this miRNA via a knockdown approach in hUC-MSCs diminished the regenerative effects of the exosomes.

#### 3.2.2. Growth Factors

The hUC, and its cells, actively secrete GFs that can potentially be used for regenerative applications, since they stimulate cell proliferation and differentiation [207]. Specifically, the hUC-MSCs are reservoirs that, in response to specific regenerative signals in their microenvironment, release molecules to promote tissue repair. In fact, Wang et al. [208] reported that intraperitoneal treatment with GM-CSFs (which are present in the hUC-MSC secretome and immunomodulate hematopoietic cells [208]) promoted follicle development (as validated through the expression of folliculogenesis-related biomarkers) [158]. Similarly, other GFs, such as HGF (enriched in the hUC-MSC secretome) [198], or EGF (loaded into collagen-based scaffolds as Matrigel^®^) [209] have also been associated with positive fertility outcomes.

#### 3.2.3. Plasma and Platelet-Rich Plasma

In terms of ovarian regeneration, Buigues et al. [61] compared the effect of hUC plasma (rich in many GFs) to that of mobilizing G-CSF treatment and demonstrated that both therapies increased the populations of growing follicles, proliferation, and angiogenesis, in addition to restoring fertility of mice with gonadotoxic damage. Furthermore, Wang et al. [210] observed the ample benefits of hUC-PRP treatment in a murine model of POI, including preserved ovarian morphology, regulated hormonal levels and estrous cyclicity, increased angiogenesis, and reduced apoptosis, with respect to the untreated groups.
ijms-23-15942-t002_Table 2Table 2Outcomes of studies applying hUC-MSCs or their derivatives (with or without bioengineered scaffolds) to treat ovarian pathologies.TreatmentModelConditionAdministrationResultsReference



Ovarian MorphologyDeveloping Follicles Serum Hormone LevelsEstrous Cyclicity Markers of Regeneration and FunctionFertility Outcomes
*hUC-MSC*RatPOITVImprovedImproved↑ E2, AMH ↓ FSHNR↑ HGF, VEGF, IGF1NR[180]*hUC-MSC*RatPOITVImprovedImproved↑ E2, P4, AMHRestored↑ Cell proliferation ↓ Apoptosis ↑ AMH, Bcl-2, FSHR ↓ Caspase-3NR[186]*hUC-MSC*RatPOITVImprovedImproved↑ E2, AMH, GnRH ↓ FSHRestored↑ Cell proliferation ↓ Apoptosis ↑ NGF, TrkA ↓ FSHR, Caspase-3NR[84]*hUC-MSC*RatPOITVImprovedImproved↑ E2, LH ↓ FSHNR↓ ApoptosisNR[182]*hUC-MSC*MousePOITVImprovedImproved↑ E2 ↓ FSHRestoredNRRestored[181]*hUC-MSC*MousePOITVImprovedImproved↑ E2, AMH ↓ FSHNR↑ Cell proliferation ↑ FSHR, Inhibin α/βRestored[187]*hUC-MSC (CD146+/−)*MousePOITVImprovedImproved↑ E2, LH ↓ FSHNR↑ Cell proliferation ↑ IL-2, TNFαRestored[185]*hUC-MSC*MousePOITVImprovedImproved↑ E2, P4 ↓ FSHNR↑ IL-4 ↓ IFNγ, NKNR[194]*hUC-MSC*MousePOITVImprovedImproved↑ E2 ↓ FSHRestoredNRNR[183]*hUC-MSC*MousePOITVNRNRNRNRNRNR[184]*hUC-MSC*RatPOILocal↑NRNRNRNRNR[188]*hUC-MSC*MousePOILocalImprovedImproved↑ E2, AMH ↓ FSHNRNRRestored[37]*hUC-MSC*MousePOILocalImprovedNR↑ E2, LH ↓ FSHNR↑ VEGFRestored[189]*hUC-MSC*HumanPOILocalImprovedImprovedNRNRNRRestored[193]*hUC-MSC*RatPOIIPNRImproved↑ E2, LH ↓ FSHNR↓ FibrosisRestored[190]*hUC-MSC*RatPOITV vs. localImprovedImproved↑ E2, AMH, LH ↓ FSHRestored↓ ApoptosisRestored[191]*hUC-MSC*RatPOITV vs. localNRImprovedNRNR↓ ApoptosisNR[36]*hUC-MSC*RatPOITV vs. localImprovedImproved↑ E2 ↓ FSHRestoredNRRestored[31]*hUC-MSC*MousePOINRNRImproved↑ E2NR↓ ApoptosisNR[35]*hUC-MSC*MouseAgingTV vs. localImprovedImproved↑ E2, P4Restored↓ Apoptosis ↓ ROS productionRestored[192]*hUC-MSC + collagen*MousePOILocalImprovedImproved↑ AMH, LH ↓ FSHRestored↑ Cell proliferation ↑ AngiogenesisNR[201]*hUC-MSC + collagen*HumanPOILocalImprovedImproved↑ E2 ↓ FSHNRNRRestored[200]*hUC-MSC + HA*MousePOILocalImprovedImprovedNRNR↓ ApoptosisRestored[197]*hUC-MSC vs. HGF*MousePOILocalNRImprovedNRNRNRNR[198]*hUC-MSC + AF*RatAbdominal adhesionsIPImprovedImprovedNRNRNRNR[196]*hUC-MSC EV*MousePOITVImprovedImproved↑ E2 ↓ FSHRestored↓ ApoptosisRestored[204]*hUC-MSC exosomes*MousePOIIPNRImproved↑ E2, AMH ↓ FSHRestored↑ Cell proliferationRestored[205]*hUC-MSC exosomes*MousePOILocalImprovedImproved↑ E2, AMH ↓ FSHNR↑ Cell proliferation ↓ Apoptosis ↓ ROS productionRestored[48]*hUC-MSC vesicles*MouseAgingLocalNRImproved↑ E2 ↓ FSHRestored↑ Oocyte qualityRestored[49]*hUC-MSC microvesicles*MousePOIVena caudalis injectionImprovedImproved↑ E2 ↓ FSHRestored↑ Angiogenesis ↑ VEGF, IGF1, Ang, AKT, p-AKTNR[107]*hUC-MSC culture medium*MousePOIIPImprovedImproved↑ AMHNRNRNR[82]*hUC-MSC + hUC-PRP*RatPOILocalImprovedNR↑ E2, AMH ↓ FSHRestored↑ Angiogenesis ↓ ApoptosisNR[210]*G-CSF vs. UC plasma*MousePOITVImprovedImprovedNRNR↑ Cell proliferation ↑ AngiogenesisRestored[61]*GM-CSF*RatPOIIPNRImprovedNRNR↑ CYP17, CD45NR[158]*EGF + Matrigel*MousePOILocalNRImprovedNRNRNRRestored[209]The up and down arrows, respectively, indicate an increase and decrease. hUC-MSC, human umbilical cord mesenchymal stem cells; POI, premature ovarian insufficiency; E2, estrogen; AMH, anti-Müllerian hormone; FSH, follicle-stimulating hormone; NR, Non-reported; HGF, hepatocyte growth factor; VEGF, vascular endothelial growth factor; IGF1, insulin growth factor 1; P4, progesterone; Bcl-2, B-cell lymphoma 2; FSHR, follicle-stimulating hormone receptor; GnRH, gonadotropin-releasing hormone; NGF, nerve growth factor; TrkA, tropomyosin receptor kinase A; LH, luteinizing hormone; hUC-PRP, human umbilical cord platelet-rich plasma; CD146, cluster of differentiation 146; IL-2, interleukin 2; TNFα, tumor necrosis factor alpha; IL-4, interleukin 4; IFNγ, interferon gamma; NK, natural killer cells; ROS, reactive oxygen species; HA, hyaluronic acid; HGF, hepatocyte growth factor; AF, amniotic fluid; EVs, extracellular vesicles; AKT, protein kinase B; p-AKT, phosphorylated-protein kinase B; Ang, angiogenin; G-CSF, granulocyte colony-stimulating factor; CYP17, cytochrome P450 17α-hydroxylase/17,20-lyase; CD45, cluster of differentiation 45; GM-CSF, granulocyte–macrophage colony-stimulating factor; EGF, endothelial growth factor; TV, tail vein injection; IP, intraperitoneal injection.


## 4. Application of Umbilical Cord Stem Cells and Their Derivatives in the Endometrium

### 4.1. Cellular Therapies Based on hUC-MSCs: Current Applications, Administration, and Fertility Restoration

There is currently a great discord apropos the gold-standard treatment for endometrial pathologies [211]. Stem cell therapies based on bone marrow-derived MSCs [212] or menstrual blood-derived MSCs [213] have been proposed. However, the beneficial effect of SCs derived from the UC on the endometrium was only reported in rats as of 2017 [39]. Table 3 summarizes the main findings of studies evaluating the potential management of endometrial pathologies by hUC-MSCs and/or hUC blood derivatives.

After endometrial scar formation, Xu et al. [39] found that hUC-MSCs recovered endometrial morphology and restored fertility in a rat model. As previously described, the SCs can be administered locally within the uterine horns, or systemically (in which case they would need to migrate from the injection site), i.e., intraperitoneally [214] or intravenous (to favor cells reaching the endometrium) [40,215]. Zhuang et al. [216] compared the efficacy of a single intravenous hUC-MSC injection with an intravenous injection combined with a local transplant. They found that treatment with hUC-MSCs promoted tissue repair (via overexpression of FGF, VEGF, HGF, and IGF1), and reduced the inflammation (by downregulating TNFα) which was more marked after combined treatment.

A promising hUC-MSC-based therapeutic strategy was evaluated in a recent clinical trial [68]. Patients with difficulty healing after uterine surgery were treated twice, with local injections of hUC-MSCs, via an 18F Foley catheter that was removed 24h later. The participants, who were followed for three and six months after treatment, reported slight improvements in menstrual blood flow, and presented reduced scarring, increased uterine volume and endometrial thickness. Notably, no adverse events were reported.

Bioengineering strategies set the foundation on which to create new biomaterials that facilitate cell adhesion into wounded areas. In this regard, several bioengineering strategies for endometrial therapies, based on hUC-MSCs, have been reported, principally using collagen scaffolds [38,39]. For example, Xin et al. [38] demonstrated that transplanting hUC-MSCs loaded in collagen scaffolds ameliorated uterine morphology, and increased cell proliferation at different time points, via the paracrine mechanisms exerted by VEGFA, TGFβ, and PDGF-BB produced by the cells. Infusions of hUC-MSCs with collagen were also reported to treat AS (characterized by the presence of intrauterine adhesions and fibrosis) and EA (characterized by suboptimal endometrial thickness that hinders embryo implantation) [67,217]. Cao et al. [217] spread the hUC-MSC-collagen complex on the patients’ uterine walls, and found the treatment restored blood flow and produced a functional endometrial within three months. Of note, almost 40% of the patients they treated achieved pregnancy by the end of a 30-month follow-up period, resulting in a total of eight healthy babies. Similarly, Zhang et al. [67] placed the collagen-driven hUC-MSCs in the uterine wall, and the procedure was performed twice. Histological analyses of biopsies confirmed collagen degradation, and hormonal replacement therapy was used to prepare for embryo transfer. Thicker endometria were observed, and ultimately, three patients achieved pregnancy following embryo transfer, and another woman got pregnant naturally.

Other scaffolds were tested in animal models to demonstrate the enhanced capacity of SC repair. Wang et al. [218] acquired human placentas, decellularized amniotic membranes, seeded them with hUC-MSCs, and used them to treat damaged rat uteri. Three estrous cycles later, the endometria had recovered normal thickness and number of glands, overexpressed metalloproteases (i.e., MMP9) and ILs (i.e., IL-4 and IL-10) related to regeneration and had lower expression of pro-inflammatory cytokines (i.e., IL-2, TNFα, and interferon gamma (IFNγ)). However, this treatment was unable to restore fertility. On the other hand, Zhou et al. [219] encapsulated the hUC-MSCs in a synthetic thermosensible hydrogel (Pluronic F-127), which was injected into rats in the right side of uterine horns nine days after they modeled a thin endometrium. This bioengineering strategy enlarged the endometrium and restored its cell proliferation and angiogenesis. Similarly, Wang et al. [220] demonstrated that treatment with hyaluronic acid loaded with hUC-MSCs successfully thickened the endometrium, increasing the number of glands and expression of pro-regenerative factors (i.e., IL-4, IGF1, and EGF), while reducing fibrosis and expression of pro-inflammatory cytokines (i.e., IFNγ), in primates with intrauterine adhesions.

Since micro-RNAs influence processes associated with tissue damage, repair, and regeneration, they may consequently be used to study endometrial regenerative activity [221]. Indeed, Sun et al. [90] found that hUC-MSCs overexpressing miR-455-5p augmented endometrial gland number and reduced fibrosis in a murine model with intrauterine adhesions. The authors argued that these regenerative processes were mediated by the activation of the JAK/STAT3 signaling pathway, which had previously been associated with cardiac subepithelial and hepatic fibrosis [222,223,224]. Alternatively, Zheng et al. [93] transfected hUC-MSCs with miR-330-5p, combined the cells with a fibroin small-intestinal submucosa scaffold, and injected them into the damaged endometrial surface. In this case, superior gland concentration was found to be mediated by the activation of the circPTP4A2-miR-330-5k-PDK2 pathway.

### 4.2. Emerging Alternatives: Acellular Therapies

#### 4.2.1. Extracellular Vesicles

Although EVs are one of the most broadly exploited bioactive agents from the hUC-MSC secretome [225], only one group has employed them for endometrial conditions [226]. Ebrahim et al. [226] assessed the therapeutic efficacy of the EVs secreted by hUC-MSCs, in a rat model of intrauterine adhesions. Endometrial function was considered to be recovered after eight weeks, with the superior number of glands, reduced fibrosis, overexpression of VEGF, and downregulation of inflammatory markers (i.e., TGFβ, TNFα, IL-1, IL-6, and RUNX Family Transcription Factor 2 (RUNX2)). Notably, this regenerative effect was amplified when the treatment was combined with estrogen.

#### 4.2.2. Growth Factors

Thus far, GF reported from hUC secretome therapy has only been tested in combination with bioengineered scaffolds in the reproductive field. Cai’s group [227] combined synthetic gelatin methacrylate (GelMA) with sodium-alginate (a natural polysaccharide) and FGF in microfluidic droplets and applied them in rats, repairing the endometrium. Interestingly, treatment using FGF combined with collagen-binding domains increased angiogenesis and endometrial thickness in rats [228], and restored menstrual blood volume, endometrial thickness, and ability to achieve pregnancy, in addition to reducing scarring, in humans [229]. Similarly, López-Martínez et al. [230] recently assessed the efficacy of a decellularized endometrial extracellular matrix hydrogel (EndoECM) loaded with a cocktail of basic FGF, IGF1, and PDGF, in a murine model of AS/EA [230]. Both groups observed an improvement in endometrial thickness and number of glands, and enhanced neoangiogenesis; however, López-Martínez et al. additionally demonstrated that the EndoECM amplified the effects of the GFs, in terms of cell proliferation and restored fertility.

#### 4.2.3. Plasma and Platelet-Rich Plasma

Plasma contains platelets which are rich in GFs that, when concentrated, can amplify their therapeutic potential in terms of tissue regeneration. Last year, De Miguel-Gómez et al. [127] reported that damaged murine uterine horns treated with UC plasma exhibited higher cell proliferation than those treated with peripheral blood plasma. Their proteomic analyses revealed overexpression of HOXA10, related to endometrial functionality, and proteins involved in angiogenesis, mitotic cell cycle, PI3K/AKT and JAK/STAT signaling pathways. Rodríguez-Eguren et al. [231] characterized thoroughly the composition of hUC-PRP processed after manual and commercial strategies. Then, its regenerative effect alone or driven by the aforementioned EndoECM was evaluated in a mouse model of AS/EA. They demonstrated the combined treatment could promote endometrial thickness, cell proliferation, angiogenesis, and restored endometrial cell function, by overexpression of AKT1, VEGF, and angiogenin. However, fertility restoration was only achieved when hUC-PRP was administered alone.
ijms-23-15942-t003_Table 3Table 3Outcomes of studies applying hUC-MSCs or their derivatives (with or without bioengineered scaffolds) to treat endometrial pathologies.TreatmentModelConditionAdministrationResultsReference



ThicknessGland NumberFibrosisRegeneration and Functionality MarkersFertility Outcomes
*hUC-MSC*RatIUATVImprovedImprovedReduced↑ Cell proliferation ↑ Angiogenesis ↑ Itga1, Thbs, Laminin, collagen ↓ VWFRestored[215]*hUC-MSC*RatIUATVImprovedImprovedReduced↑ Cell proliferation ↑ Angiogenesis ↑ VEGFA, MMP9, CD31 ↓TNFα, IFNγ, IL-2, IL-4, IL-10Restored[40]*hUC-MSC*RatIUALocalImprovedImprovedNRNRRestored[232]*hUC-MSC*RatThin endometriumTV + LocalNRNRNR↑ FGF ↓ TNFαNR[216]*hUC-MSC*RatIUAIPImprovedImprovedReduced↑ Angiogenesis ↓ TGFβ and Smad3Restored[214]*hUC-MSC*HumanIUALocalImprovedNRReducedRestored menstrual cycleNR[68]*hUC-MSC + collagen*RatIUALocalImprovedImprovedReduced↑ Angiogenesis ↑ MMP9Restored[39]*hUC-MSC + collagen*HumanIUALocalImprovedNRReduced↑ Cell proliferation ↑ PanCK, ERα, PR ↑ VEGFA, TGFβ, PDGFRestored[38]*hUC-MSC + collagen*HumanAsherman syndromeLocalImprovedNRReduced↑ Cell proliferation ↑ Angiogenesis ↑ ERα, PRRestored[67]*hUC-MSC + collagen*HumanIUALocalImprovedNRReduced↑ Cell proliferation ↑ Angiogenesis ↑ ERα, PRRestored[217]*hUC-MSC + AMM*RatIUALocalImprovedImprovedNR↑ Keratin, Vimentin, Integrinβ3, IL-4, IL-10, MMP9, KI67 ↓TNFα, IFNγ, IL-2, VEGFNon Restored[218]*hUC-MSC + PF-127*RatThin endometriumLocalImprovedImprovedNR↑ Cell proliferation ↑ Angiogenesis ↑ VEGFA, Nos3NR[219]*hUC-MSC + SF-SIS*MouseIUALocalImprovedImprovedReducedNRNR[93]*hUC-MSC + HA*MonkeyIUALocalImprovedImprovedReduced↑ IL-4, IGF1, EGF ↓ IFNγNR[220]*hUC-MSC^miR−455−5p^*MouseIUANRNRImprovedReduced↑ JAK2, STAT3 ↓ SOCS3NR[90]*hUC-MSC EVs*RatIUAIPNRImprovedReduced↑ VEGF ↓ TGFβ, TNFα, IL-1, IL-6, RUNX2, COL1A1NR[226]*UC plasma*MouseIUALocalNRNRNR↑ Cell proliferation ↑ HOXA10, P85, 2aaa, Stat5A, RhoaNR[127]*hUC-PRP + EndoECM*MouseIUALocalImprovedImprovedReduced↑ Cell proliferation ↑ Angiogenesis ↑ AKT1, VEGF, angiogeninRestored[231]*PDGF-BB + FGF + IGF1 + EndoECM*MouseIUALocalImprovedImprovedReduced↑ Cell proliferation ↑ Angiogenesis ↓ Col1A1Restored[230]*FGF + CBD*RatIUALocalImprovedNRReduced↑ AngiogenesisRestored[228]*FGF + CBD*HumanIUALocalImprovedNRReduced↑ Cell proliferation ↑ AngiogenesisRestored[229]*FGF + GelMA + Na-alginate scaffold*RatIUALocalImprovedImprovedReduced↑ AngiogenesisNR[227]The up and down arrows, respectively, indicate an increase and decrease. hUC-MSC, human umbilical cord mesenchymal stem cells; IUA, intrauterine adhesions; Itga1, integrin subunit alpha 1; Thbs, thrombospondin; VWF, Von Willebrand factor; VEGF, vascular endothelial growth factor, VEGFA, vascular endothelial growth factor A; MMP9, metalloprotease 9; CD31, Cluster of differentiation 31; TNFα, tumor necrosis factor alpha; IFNγ, interferon gamma; IL-2, interleukin 2; IL-4, interleukin 4; IL-10, interleukin 10; NR, non-reported; FGF, fibroblast growth factor; TGFβ, transforming growth factor beta; Smad3, SMAD family member 3; PanCK, pan-cytokeratin; ERα, estrogen receptor alpha; PR, progesterone receptor; PDGF, platelet-derived growth factor; AMM, acellular amniotic matrix; PF-127, poly(ethylene oxide)-poly(propylene oxide)-poly(ethylene oxide) 127; Nos3, nitric oxide synthase 3; SF-SIS, silk fibroin small-intestinal submucosa; HA, hyaluronic acid; IGF1, insulin-like growth factor 1; EGF, epidermal growth factor; JAK2, Janus kinase 2; STAT3, signal transducer and activator of transcription 3; SOCS3, suppressor of cytokine signaling 3; miR-455-5p: microRNA 455-5p; EVs: extracellular vesicles; IL-1, interleukin 1; IL-6, interleukin 6; RUNX2, runt-related transcription factor 2; COL1, collagen type 1; HOXA10, homeobox A10; 2aaa, serine/threonine-protein phosphatase 2A; Stat5A, signal transducer and activator of transcription 5A; Rhoa, Ras homolog family member A; CBD, collagen-binding domain; GelMA, gelatin methacrylate; TV, tail vein injection; IP, intraperitoneal injection; EndoECM, decellularized endometrial extracellular matrix hydrogel; UC, umbilical cord; hUC-PRP, human umbilical cord platelet-rich plasma; Na-alginate, sodium alginate.


## 5. Applications of Umbilical Cord Stem Cells and Their Derivatives in Other Female Reproductive Organs

### 5.1. Vagina

Vaginal pelvic organ prolapse is a condition in which one or more organs in the pelvis slip down from their original position and protrude into the vagina [233]. Diverse SC therapies have achieved promising results in preclinical studies, conferring tissue elasticity, muscle regeneration, and reduced inflammatory reactions [234]. Specifically, a rhesus macaque model of pelvic organ prolapse treated with hUC-MSCs showed increased muscular thickness, higher elasticity of the vaginal wall, upregulation of muscular (i.e., COL1A1 and FBN5) and neovascular markers (e.g., VEGF), and downregulation of matrix metalloproteinases (i.e., MMP2/9/13). The innovative strategy of loading hUC-MSCs into synthetic scaffolds was also reported using the same primate model. A scaffold derived from the small intestine submucosa was employed to deliver the hUC-MSCs, which improved angiogenesis, muscular thickness, collagen content, and vaginal wall elasticity, in addition to upregulating muscular markers (i.e., COLI, ACTA, and ELN) and metalloproteases (i.e., MMP1/2), and reorganized the extracellular matrix [235]. Moreover, the use of a collagen I scaffold in a rat model improved the collagen content in the vaginal wall and vaginal elasticity, in addition to upregulating ACTA2, Elastin, FGF, vimentin, COL3A1, TIMP metallopeptidase inhibitor 1 (TIMP1), VEGFA, and TGFβ [236]. Similarly, the use of a synthetic polypropylene scaffold to deliver the hUC-MSCs decreased the inflammatory response, upregulated IL-1, -4, and -10, and downregulated MMP1/9 [237].

We highlight that, so far, only one group has applied hUC-MSCs in humans for vaginal diseases. This particular study was related to stress urinary incontinence (where the urine leaks without bladder contraction), which affects 10–40% of women [238,239]. The hUC-MSCs were injected into the submucosal area of the proximal-urethra, in a two-step procedure, and no adverse events were reported. After measuring urodynamic markers, Lee et al. [240] demonstrated the deficient intrinsic sphincter and mixed stress incontinency were improved by this treatment.

### 5.2. Oviducts

The only application of hUC-MSC within the oviducts was reported in 2019, for chronic salpingitis caused by *Chlamydia trachomatis* [241]. Chronic salpingitis compromises the oviductal structure, leading to distal tube obstruction and hydrosalpinx, which are well-known causes of infertility [242]. Liao et al. [241] pioneered this work by isolating MSCs from UC donors, labelling, and transplanting them into chlamydial-infected mice. Within four weeks, the SC therapy had improved fallopian tube morphology, increased the proliferation index, and reduced apoptosis and inflammation, compared to the non-treated group. Individuals treated with the hUC-MSCs were also capable of achieving pregnancy and live births.

### 5.3. Placenta

Spontaneous abortion is defined as the loss of a pregnancy, without outside intervention, during the first 20 weeks of gestation [242]. The consecutive loss of three or more pregnancies is diagnosed as recurrent pregnancy loss, and may be caused by uterine anatomical defects, chromosomal aberration, or immunological abnormalities [243]. The therapeutic potential of hUC-MSCs in abortion was studied using a rat model generated by Chen et al. [244]. This group induced abortion with bromocriptine, between gestational days 6–8, and transplanted the hUC-MSCs on day 9. This treatment prevented necrosis of the placental decidual cells (usually provoked by abortion), and restored levels of IL-10, -17, and IFNγ to normal, in addition to reducing early pregnancy loss after mating. However, the long-term safety and effectiveness of this treatment merits further investigation.

## 6. Pros and Cons of Using Human Umbilical Cord Stem Cells and Their Derivatives

Despite the application of hUC-MSCs proving to have therapeutic benefits for reproductive diseases in preclinical models, further studies are required to assess their efficacy in the clinical setting and long-term safety (Figure 2). Once they reach their target tissue, the hUC-MSCs can differentiate into resident cells (depending on the signals they receive from the tissue-specific ECM), and thus potentially provide a lifetime supply of cells for the patient, with only a single dose. However, SC therapy has some limitations, including the need to establish a bank of hUC-MSCs, and their tumorigenic potential, which makes clinical translation challenging [245].

Acellular therapies based on the hUC-MSC secretome, or specific hUC blood derivatives, are emerging as promising alternatives in reproductive medicine. Specifically, EVs and hUC-MSCs are postulated to have similar biological effects, but EVs are more advantageous in terms of greater stability, easier storage, diminishing the risk of ectopic tissue formation, possible immune rejection, and tumorigenicity, when compared to cell transplantation (Figure 2) [110]. Similarly, the acellular components of hUC blood, such as hUC plasma, hUC-PRP, and GFs, exhibited lower immunogenicity and tumorigenicity, compared to cell transplantation. Apart from being more economical in the long-term [245], these components are microbiologically safe, and avoid the extra burdens associated with autologous treatments [137]. Furthermore, soluble GFs in the hUCS and hUCP remain stable after extended periods of cryopreservation, facilitating their management [246]. Given these reasons, we expect that the acellular components of human UC blood—particularly PRP—will be key players in the future of regenerative medicine. However, future studies should be conducted to develop strategies that mitigate potential problems with their targeting or retention [225].

Advances in bioengineering offer new methodologies to sustain delivery of cells and paracrine factors, boosting efficiency and maintaining long-term efficacy [199]. Thus far, only collagen-derived scaffolds have been applied in clinical trials, for patients affected by POI [200] and AS [67,217]. However, tissue-specific hydrogel scaffolds that mimic the target microenvironment are quickly emerging as alternatives, since they potentiate the restorative effects [247], as has been demonstrated with murine models of AS [230].

Remarkably, the US Food and Drug Administration (FDA) states the products used in regenerative medicine must be regulated prior to their implementation in the industry and into patients. Specifically, the safety, purity, potency, and identity of the hUC blood or their derivatives, as well as their testing in clinical trials, are mandatory [248].

Due to the lack of studies comparing cellular and acellular therapies, it is difficult to evaluate which of the therapies described herein is superior. Nevertheless, the ongoing clinical trials assessing the application of hUC-PRP in the endometrium [138], and hUC-MSCs in the endometrium [249,250] and ovaries [251,252,253], bring these products one step closer to clinical translation and, ultimately, helping to treat infertility-related conditions. Further information about these ongoing clinical trials is provided in Appendix A.

Notably, treatments based on hUC blood components are also emerging as alternatives to treat male infertility. Preclinical models have already been used to demonstrate the ability of hUC-MSCs to differentiate into germ cells, in the lumen of the seminiferous tubules [254,255], and their therapeutic effect against the gonadotoxicities of lead [256]. However, research in this field remains experimental, as there have not been any reports of clinical trials evaluating their use to treat male infertility-related conditions [257].

## 7. Conclusions

In conclusion, regenerative medicine targeting female reproduction has generated promising outcomes through the use of hUC-MSCs and blood derivatives, such as hUC-MSCs, hUC-PRP, and EVs, in the treatment of multiple gynecological disorders (i.e., POI, AS, EA, pelvic organ prolapse, or abortions). These treatments enhanced tissue regeneration by upregulating the secretion of GFs and cytokines and through their immunomodulatory actions. Further studies are required to establish a clinical benefit, and to determine whether cellular or acellular therapy is superior.

## Figures and Tables

**Figure 1 ijms-23-15942-f001:**
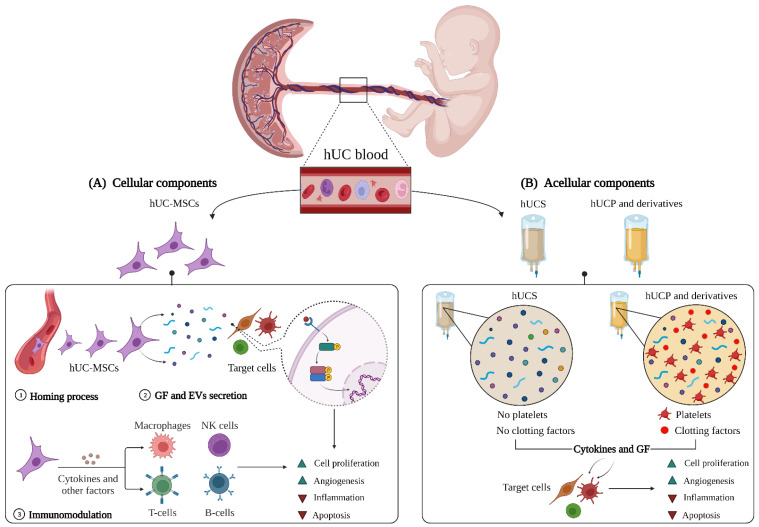
The hUC blood composition. This illustration depicts the roles of the (**A**) cellular and (**B**) acellular components, in tissue regeneration processes. EVs, extracellular vesicles; GF, growth factors; hUC, human umbilical cord; hUC-MSCs, human umbilical cord mesenchymal stem cells; hUCP, human umbilical cord plasma; hUCS, human umbilical cord serum; and NK cells, natural killer cells.

**Figure 2 ijms-23-15942-f002:**
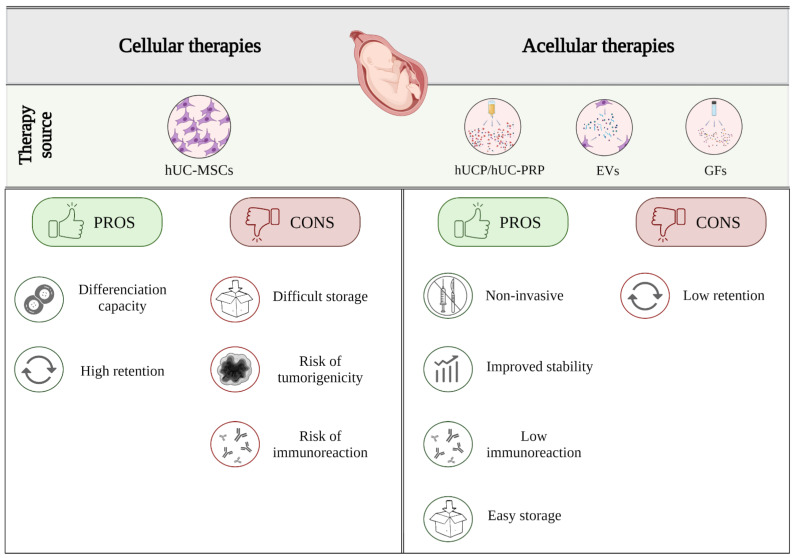
Pros and cons of using cellular and acellular UC therapies in the female reproductive tract. hUC-MSCs: human umbilical cord mesenchymal stem cells; hUCP: human umbilical cord plasma; hUC-PRP: human umbilical cord platelet-rich plasma; EVs: extracellular vesicles; GFs: growth factors.

## Data Availability

Not applicable.

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
