# Peer review of "Human Umbilical Cord-Based Therapeutics: Stem Cells and Blood Derivatives for Female Reproductive Medicine"

_ijms, 2022, doi:10.3390/ijms232415942_

Round 1

Reviewer 1 Report

The review by Rodríguez-Eguren et al, describes the available therapeutic options for female reproductive disorders using human umbilical cord blood derivatives. It is an overall well-written review with comprehensive coverage of pre-clinical research on use of hUC and MSC for treatment of human diseases.

Following are the comments:

1.       Move the following appendices for the tables to the end of the text:

Line 275-280, Line 405-416, Line 524-539,

2.       The content of Table 1 describing human diseases and not female reproductive disorders, does not fit into the scope of the review about treatment of reproductive disorders. I am not sure that it adds value to the review.

3.       Table 2 and 3

Review and correct the format of titles to make it easily readable

4.       In Table 2: In columns Ovarian morphology, developing follicles- describe the findings in few words. The arrows aren’t intuitive.

5.       Are there any clinical trials using hUC for infertility treatments? If so, please include a table describing those.

6.       I suggest making a table describing pros and cons if they can be easily contrasted. *Suggested- not required.

Author Response

Please, find attached the answers to the reviewer.

Reviewer 2 Report

The manuscript by Eguren et. al, provides a comprehensive review of literature about human umbilical cord mesenchymal stem cells (hUC-MSC) along with their secreted paracrine factors and biomolecules as promising therapeutics in regenerative medicine. Overall, the manuscript is well written and will be very well-suited for publication in IJMS once the following comments are addressed:

1.     Both the figures (Figure 1 and Figure 2) are poorly presented and convey very less information about the subject and needs to be improved.

2.     The addition of a short summary regarding FDA approved cord blood treatments would strengthen the manuscript and authors conclusions. 

Author Response

Please, find the responses to the queries and comments from the reviewer.
